# Immune Analysis Using Vitreous Optical Coherence Tomography Imaging in Rats with Steroid-Induced Glaucoma

**DOI:** 10.3390/biomedicines12030633

**Published:** 2024-03-13

**Authors:** Maria J. Rodrigo, Manuel Subías, Alberto Montolío, Teresa Martínez-Rincón, Alba Aragón-Navas, Irene Bravo-Osuna, Luis E. Pablo, Jose Cegoñino, Rocío Herrero-Vanrell, Elena Garcia-Martin, Amaya Pérez del Palomar

**Affiliations:** 1Department of Ophthalmology, Miguel Servet University Hospital, 50009 Zaragoza, Spain; mariajesusrodrigo@hotmail.es (M.J.R.); manusubias@gmail.com (M.S.); teresamrincon@gmail.com (T.M.-R.); lpablo@unizar.es (L.E.P.); 2Miguel Servet Ophthalmology Research Group (GIMSO), Aragon Health Research Institute (IIS Aragon), University of Zaragoza, 50009 Zaragoza, Spain; 3National Ocular Researcha Network RD21/0002/0050, RICORS Red de Enfermedades Inflamatorias (RD21/0002), Carlos III Health Institute, 28220 Madrid, Spain; 4Biotech Vision, Instituto Oftalmologico Quiron, 50012 Zaragoza, Spain; 5Biomaterials Group, Aragon Engineering Research Institute (I3A), University of Zaragoza, 50018 Zaragoza, Spain; amontolio@unizar.es (A.M.); jcegoni@unizar.es (J.C.); amaya@unizar.es (A.P.d.P.); 6Department of Mechanical Engineering, University of Zaragoza, 50018 Zaragoza, Spain; 7Innovation, Therapy and Pharmaceutical Development in Ophthalmology (InnOftal) Research Group, UCM 920415, Department of Pharmaceutics and Food Technology, Faculty of Pharmacy, Complutense University of Madrid, 28040 Madrid, Spain; albarago@ucm.es (A.A.-N.); ibravo@ucm.es (I.B.-O.); rociohv@ucm.es (R.H.-V.); 8Health Research Institute of the San Carlos Clinical Hospital (IdISSC), 28040 Madrid, Spain; 9University Institute of Industrial Pharmacy (IUFI), School of Pharmacy, Complutense University of Madrid, 28040 Madrid, Spain

**Keywords:** optical coherence tomography, vitreous body, glaucoma, animal models, inflammation

## Abstract

Glaucoma is a multifactorial pathology involving the immune system. The subclinical immune response plays a homeostatic role in healthy situations, but in pathological situations, it produces imbalances. Optical coherence tomography detects immune cells in the vitreous as hyperreflective opacities and these are subsequently characterised by computational analysis. This study monitors the changes in immunity in the vitreous in two steroid-induced glaucoma (SIG) animal models created with drug delivery systems (microspheres loaded with dexamethasone and dexamethasone/fibronectin), comparing both sexes and healthy controls over six months. SIG eyes tended to present greater intensity and a higher number of vitreous opacities (*p* < 0.05), with dynamic fluctuations in the percentage of isolated cells (10 µm^2^), non-activated cells (10–50 µm^2^), activated cells (50–250 µm^2^) and cell complexes (>250 µm^2^). Both SIG models presented an anti-inflammatory profile, with non-activated cells being the largest population in this study. However, smaller opacities (isolated cells) seemed to be the first responder to noxa since they were the most rounded (recruitment), coinciding with peak intraocular pressure increase, and showed the highest mean Intensity (intracellular machinery), even in the contralateral eye, and a major change in orientation (motility). Studying the features of hyperreflective opacities in the vitreous using OCT could be a useful biomarker of glaucoma.

## 1. Introduction

Chronic glaucoma is a leading cause of irreversible blindness in the world [1]. Increase in intraocular pressure (IOP) is a risk factor strongly associated with the onset and progression of this optic neuropathy. However, several studies have indicated that the pathogenesis of the disease is multifactorial, and the immune perspective seems to be of great relevance [2]. Residential glial cells are found to become activated in the early stages of glaucoma. Elevated IOP triggers secondary responses responsible for retinal ganglion cell (RGC) degeneration. Although the primary response may be favourable in protecting the eye, the subsequent events that lead to long-lasting activation of glial cells and adaptive immune responses can be destructive [3]. RGC death results in irreversible visual field impairment [4,5] that is only detectable once 25–30% of RGCs is lost, leading to delayed diagnosis. It is therefore essential to develop new tools and markers to enable earlier detection. Furthermore, the association between autoimmunity and progressive neuron loss in glaucoma may also allow the development of novel therapeutic interventions that eventually offer a cure for the disease.

Immune cells present different morphologies based on their state of activation [6,7,8]. Soma size, analysed by in vivo fluorescence imaging, was proposed as a significant marker of immune activation in the brain [6] and retina of glaucomatous mice [9]. Microglial activation (Iba1+ staining) appears to be the earliest detectable change in the retina [10] that strongly correlates with and predicts the severity of glaucomatous neurodegeneration [11]. Very few studies, however, have extensively analysed the vitreous in entities with parainflammation [12]. Hyalocytes [13] are resident vitreous cells that participate in immune regulation by means of phagocytic activity and their contractile properties. In response to noxa, they are replaced and increase their mitotic activity. All these changes are postulated as early biomarkers of value for diagnosing ocular diseases [14].

Optical coherence tomography (OCT) is an objective, fast and cost-efficient technology that allows in vivo acquisition of high-resolution cross-sectional images micrometres from the eye structures. Latest-generation OCT systems allow non-invasive evaluation of the vitreous in acute and chronic inflammatory processes under standard clinical conditions. They also allow evaluation of the changes that occur after treatment [15,16,17]. In our previous paper on the use of computational OCT image analysis, we demonstrated that hyalocyte-like Iba1+ cells were observed as hyperreflective opacities and described their behaviour in the active/non-active state by characterising them in terms of size, intensity, eccentricity and orientation in two chronic glaucoma models in rats with ocular hypertension (OHT) [18].

This paper aims to corroborate the reliability of using computational OCT image analysis of hyperreflective opacities in the vitreous as a biomarker of vitreous immunity, in this case in two chronic steroid-induced glaucoma (SIG) rat models previously developed by our research group by injecting biodegradable microspheres (Ms) loaded with dexamethasone (MsDx) and a combination of dexamethasone and fibronectin (MsDxF) (with sustained release of the active compounds) into the anterior chamber of the eye [19,20]. Chronic exposure to glucocorticoids can raise IOP and is known to exert a negative effect in the form of maladaptive glial cell alterations and neuron damage or loss [21], leading to SIG [22]. We corroborate and validate the computational analysis of the individual hyperreflective opacities as a better technique than the overall relative measure of immunity using OCT. The study of eccentricity, intensity and orientation characteristics of vitreous opacities using OCT is a reproducible and reliable method of non-invasive assessment of SIG.

## 2. Materials and Methods

### 2.1. Data Collection

The dataset comprised images of the vitreoretinal interface obtained using OCT (HR-OCT Spectralis, Heidelberg^®^ Engineering, Heidelberg, Germany) in two previous interventional studies on the generation of steroid-induced glaucoma models (MsDx and MsDxF) [19,20], which detail the methodology followed. The experiment was previously approved by the Ethics Committee for Animal Research (PI34/17) of the University of Zaragoza (Spain) and was carried out in strict accordance with the Association for Research in Vision and Ophthalmology’s Statement on the Use of Animals. The MsDx model was generated by injecting a 2-microlitre suspension (10% w) of biodegradable PLGA microspheres [23] loaded with dexamethasone into the anterior chamber of Long–Evans rats’ right eyes at 0 and 4 weeks [19]. The second model (MsDxF) was generated by administering a 2-microlitre suspension (10% w) of biodegradable PLGA microspheres co-loaded with dexamethasone and fibronectin at baseline in a single injection [20]. The left eyes did not undergo intervention. IOP (using a Tonolab^®^ rebound tonometer) measurement and OCT scans of both eyes were performed at 0, 2, 4, 6, 8, 12, 18 and 24 weeks. A cohort that did not undergo intervention served as the control and was scanned at 0, 12 and 24 weeks.

### 2.2. Image Analysis

Images were acquired using a high-resolution OCT device with a plane power polymethylmethacrylate contact lens (thickness 270 μm, diameter 5.2 mm) (Cantor+Nissel^®^, Northamptonshire, Northampton, UK) adapted to the rat cornea [24]. The retinal posterior pole protocol with automatic segmentation, eye-tracking software and a tracking application were used to ensure that the same points were re-scanned throughout this study. “Enhance depth imaging” mode was disabled in all cases.

The raw OCT images were exported in Audio Video Interleave (AVI) format. In the rodent version of this OCT device, the videos were composed of cross-sectional images acquired from 61 3 mm long B-scans centred on the optic nerve. These cross-sectional images had a resolution of 3 μm/pixel and an area of 2.906 mm^2^ (1536 × 496 pixels). Therefore, each pixel had an area of 3.815 µm^2^. These videos were analysed using a custom program implemented in MatLab (version R218a, MathWorks Inc., Natick, MA, USA). The imaging data were analysed by a masked reader. Two different researchers, likewise masked, performed OCT segmentation to verify reproducibility.

In order to measure the immune response, relative intensity in the vitreous/retinal pigment epithelium (VIT/RPE) was quantified [15,25,26]. Our customised program segments the vitreous and RPE by locating the inner limiting membrane (ILM) and the inner and outer layers of the RPE using greyscale conversion (Figure 1). VIT/RPE intensity was calculated as the mean of the pixel intensity in each region, giving VIT/RPE relative intensity in each cross-sectional image. VIT/RPE relative intensity in each eye is the mean of all B-scans.

The vitreous opacities in each cross-sectional image were analysed as they are closely related to the immune cells. OCT analysis of hyperreflective opacities in the vitreoretinal interface does not require a correction factor for histological correlation [27] and ensures the characterisation of the actual opacity. These opacities were classified according to size based on previous morphological analyses of retinal microglia and histological analyses of hyalocytes [28]. Soma size can be used to discriminate between non-activated and activated cells, as the morphology of microglia varies according to their state of activation: the smallest cells (corresponding to early growth) have a rounded or amoeboid morphology; resting (non-activated) cells have a thin cell body with branched cellular processes; and reactive (activated) cells have a larger somatic size and exhibit phagocytic activity and motility [6,7].

Our custom program automatically measured hyperreflective opacities and classified them into groups according to their size: isolated cells (<10 μm^2^), non-activated cells (10–50 μm^2^), activated cells (50–250 μm^2^) and cell complexes (>250 μm^2^). The size of the opacities was calculated according to the number of pixels in each opacity. Background intensity is lower than opacity intensity; therefore, background speckle noise was removed to ensure the measurement of hyperreflective opacities (see Figure 1). In this way, the physiological ocular phenomena were eliminated [29].

Several parameters can be calculated for each opacity. The total cell area, calculated by the number of opacities and the area of each opacity, represented the overall immune response to the induced glaucoma model. The mean number of opacities was an indicator of immunity to noxa over time, allowing analysis of in situ resident immune cellularity and intra- or extra-ocular recruitment [30,31,32,33]. The mean area of opacities was calculated for all cells and for each group according to cell size, attaining reliable cell soma reproducibility. The changing proportion between the activated and non-activated cell populations was analysed by quantifying the cell percentage for each group.

Opacity/cell intensity, calculated as the mean of the intensity of each pixel in the opacity, is related to immune activation because it implies gene–protein expression prior to soma remodelling. Eccentricity was also calculated: values close to 1 indicate linear, elongated or flat cell morphology, while values close to 0 represent a rounded shape. Opacity/cell orientation was used as an indirect parameter of motility or active displacement of immunity towards the damage [7,9,11,34,35].

### 2.3. Statistical Analysis

All data were recorded in an Excel database and statistical analysis was performed using SPSS software version 20.0 (SPSS Inc., Chicago, IL, USA). The variables under study were eyes (intervened right eye versus non-intervened left eye), sex (male versus female), type of steroid-induced glaucoma model (MsDx versus MsDxF) and control, number of injections, IOP and vitreous signal features using OCT (VIT/RPE relative intensity, total area, mean number of opacities, mean area of opacities, opacity percentage and opacity eccentricity, intensity and orientation).

After checking for variable normality with the Kolmogorov–Smirnov test, we performed a parametric test using multiple ANOVA comparisons and correlations with Pearson’s P test. All values were expressed as mean ± standard deviations. Values of *p* < 0.05 were considered to indicate statistical significance, and the Bonferroni correction for multiple comparisons was calculated to avoid a high false-positive rate. In Figure 2 and Figure 3, statistically significant differences are indicated as follows: A (MsDx–MsDxF), B (MsDx–control), C (MsDxF–control). Figure 4, Figure 5, Figure 6, Figure 7, Figure 8 and Figure 9 show isolated cells (<10 µm^2^; group 1), non-activated cells (10–50 µm^2^; group 2), activated cells (50–250 µm^2^; group 3) and cell complexes (>250 µm^2^; group 4). Statistically significant differences (*p* < 0.05) are indicated with alphabetic markers as follows: a (group 1–group 2), b (group 1–group 3), c (group 1–group 4), d (group 2–group 3), e (group 2–group 4) and f (group 3–group 4).

## 3. Results

### 3.1. Microsphere Characterisation

Both microsphere formulations (MsDx and MsDxF) were spherical and had a mean particle size of approximately 14 µm and a unimodal particle size distribution. The microspheres’ surface was influenced by the production method: those prepared via evaporation solvent from a simple emulsion (MsDx) had non-porous surfaces, according to scanning electron microscopy (SEM), while the use of the double-emulsion technique (MsDxF) produced small surface pores in the microspheres. Dexamethasone loading was approximately 60 µg DX/mg Ms for the MsDx and approximately 72 µg DX/mg Ms for the MsDxF. In both cases, sustained release of the active compounds was observed for several weeks. For a more detailed description of these results, see previous studies published by the research group [19,20,36].

### 3.2. Ophthalmological Analysis

A total of 280 OCT videos, obtained from 120 rats (60% females/40% males) at different times of study follow-up, were analysed. MsDx (n = 43 rats): 49 videos from the right eye (RE)/49 videos from the left eye (LE); MsDxF (n = 44): 44 RE/50 LE; healthy controls (n = 32): 31 RE/57 LE. IOP progressively increased in both SIG models and differences were found between the sexes. Glaucomatous and healthy males had higher IOP levels than females throughout the study (data extracted from [19,20,37]) (Figure 2).

**Figure 2 biomedicines-12-00633-f002:**
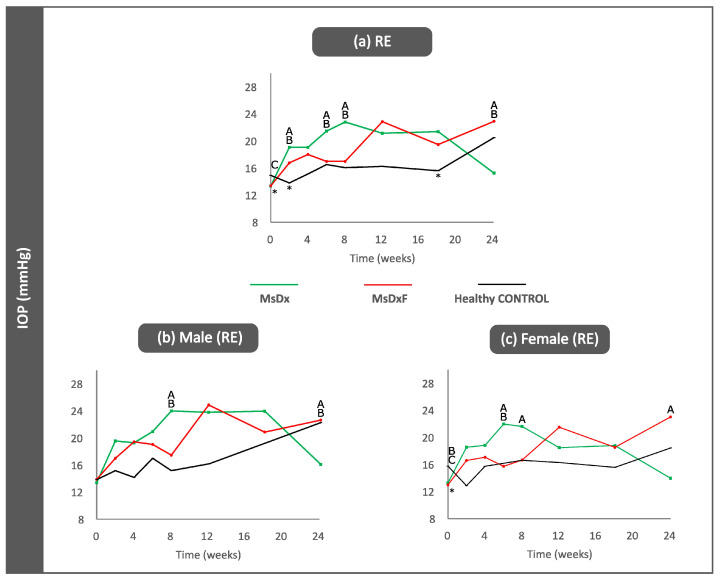
Intraocular pressure curves (right eyes) in two steroid-induced glaucoma models and healthy controls for all right eyes (**a**), for the right eyes of the males (**b**) and for the right eyes of the females (**c**). Abbreviations: MsDx: cohort with microspheres loaded with dexamethasone; MsDxF: cohort with microspheres loaded with dexamethasone and fibronectin injected into the anterior chamber; IOP: intraocular pressure (data extracted from [37,38]). *: statistical significance (*p* < 0.05) between glaucoma models and healthy controls (ANOVA); A: significant differences between MsDx and MsDxF; B: significant differences between MsDx and healthy controls; C: significant differences between MsDxF and healthy controls.

### 3.3. Computational Analysis

#### 3.3.1. VIT/RPE Intensity

OCT analysis of the vitreous detected slightly higher VIT/RPE intensities in the injected right eyes in both SIG models in the final stages of this study (*p* > 0.05). The MsDx model generated with two injections showed a higher VIT/RPE signal than the MsDxF model generated with a single injection throughout the study; however, after the first injection (week 2), both SIG models showed similar VIT/RPE intensities to healthy controls (Figure 3a). Non-injected left eyes showed a slight increase in vitreous signal intensity versus healthy controls (Figure 3b). Healthy control animals’ IOP and vitreous signal intensity measurements were lower than those of both SIG animals (Figure 2 and Figure 3). Lastly, the influence of sex was analysed. In general, females with SIG showed slightly higher VIT/RPE OCT intensity than males and their healthy female counterparts. Healthy control males showed increased vitreous intensity at week 12 (16 weeks of life), after which vitreous intensity declined (Figure 3c,d).

**Figure 3 biomedicines-12-00633-f003:**
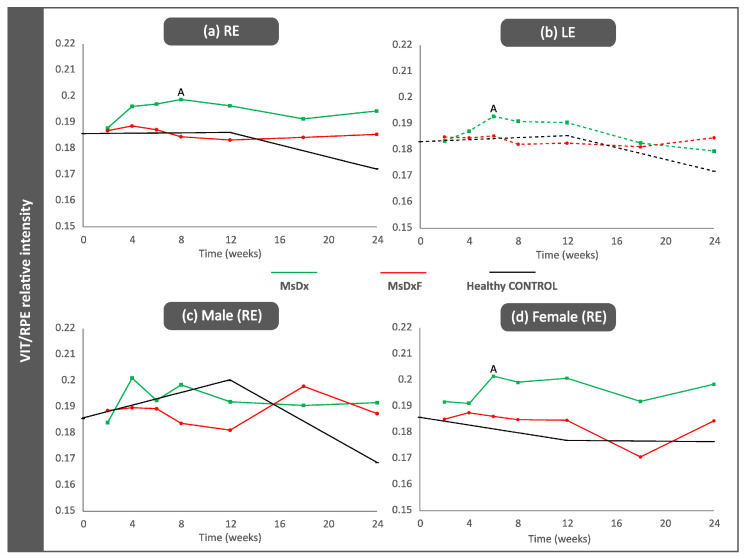
VIT/RPE signal intensity. (**a**) Right eye from both sexes; (**b**) Left eye from both sexes; (**c**) males; (**d**) females. Abbreviations: RE: right eye; LE: left eye; MsDx: cohort with microspheres loaded with dexamethasone (green); MsDxF: cohort with microspheres loaded with dexamethasone and fibronectin injected into the anterior chamber (red); healthy CONTROL: cohort of healthy animals non intervented (black); VIT: vitreous; RPE: retinal pigment epithelium. A: significant differences between MsDx and MsDxF.

#### 3.3.2. Correlation Analysis

A correlation study was performed with the aim of evaluating the influence of the model on the VIT/RPE intensity. Both SIG models are induced by intraocular injections, which involve rupture of the eye barrier and induce anterior chamber-associated immune deviation (ACAID) [39,40]. However, no strong statistically significant correlations between either injections or intensities analysed by OCT were found in any SIG model or in the healthy cohort, which implies a lower level of immune involvement. The most relevant results and correlations are shown in bold in Table 1.

MsDx cohort: Both sexes presented an inverse correlation between IOPs at different times, which was moderate in females (IOP 0 w/6 w in the right eye and IOP 2 w/18 w in the left eye) and strong in males (IOP 2 w/12 w; r = −0.825, *p* = 0.012). Eyes with initially lower IOPs were more likely to present higher IOPs at later times [18]. Furthermore, IOP at early stages (2 and 4 w) correlated directly with OCT intensity at the final stages (24 w in the right eye and 18 w in the left eye). This suggests a greater anti-inflammatory effect exerted by dexamethasone in the injected right eye, delaying the IOP/OCT intensity correlation in both sexes (males IOP 2 w/OCT 24 w; r = 0.999, *p* = 0.029, and females IOP 8 w/OCT 18 w; r = 0.999, *p* = 0.012). The inverse correlation (possibly reflecting the protective effect of dexamethasone) was observed in males at week 6 (IOP 6 w/OCT 6 w; r = −0.999, *p* = 0.030) and in females at week 8 (IOP 2 w/OCT 8 w; r = −0.999, *p* = 0.019). This protection was lost later when a direct correlation was found in males at week 8 (IOP 8 w/OCT 8 w; r = 0.999, *p* = 0.028) and in females at week 18 (IOP 8 w/OCT 18 w; r = 0.999, *p* = 0.012). This suggests earlier loss of anti-inflammatory action due to dexamethasone in males since a direct IOP/OCT correlation was found earlier in this sex.

MsDxF cohort: Females also presented a moderate inverse correlation (IOP 0 w/18 w) and a direct correlation (IOP 0 w/12 w), supporting the more progressive and delayed IOP increase in right eyes. This was also the case in left eyes (IOP 4 w/24 w; r = 0.919, *p* = 0.010). However, males showed a strong direct correlation earlier (IOP 2 w/8 w; r = 0.841, *p* = 0.002), which seems to imply a predisposition for an earlier IOP increase in males in this model. No IOP/OCT correlations were found in the injected right eye. However, male left eyes showed a strong inverse correlation at 12 w (IOP 0 w/OCT 12 w; r = −0.851, *p* = 0.032) and a direct correlation at 8 w (IOP 0 w/OCT 8 w; r = 0.882, *p* = 0.020, and IOP 6 w/OCT 8 w; r = 0.813, *p* = 0.049) and at 18 w (IOP 2 w/OCT 18 w; r = 0.999, *p* = 0.022).

Healthy control cohort: An early and positive IOP correlation (IOP 4 w/IOP 8 w; r = 0.934, *p* = 0.020) was observed in both sexes. Females showed an inverse correlation according to IOP at early stages (IOP 0 w/6 w; r = −0.999, *p* = 0.021) and a direct IOP/OCT correlation at the intermediate (IOP 12 w/OCT 12 w; r = 0.997, *p* = 0.049, in the left eye) and late stages (IOP 18 w/OCT 24 w; r = 0.854, *p* = 0.031, in the right eye). In females, the age-related degenerative process [41,42] produces higher vitreous OCT intensity (reflex of immune involvement and/or activation) correlated with ocular normotension. But males showed a moderate inverse correlation at the end of this study (IOP 24 w/OCT 24 w).

#### 3.3.3. In Vivo Analysis of Vitreous Immunity

In our previous paper, we showed that the hyperreflective opacities in the vitreous corresponded to hyalocyte-like Iba1+ cells [18,43] and that hypertensive eyes revealed many hyalocyte-like cells surrounding the ciliary body, some of which migrated from the ciliary body, crossing to the vitreous cavity [14]. In this study, the characteristics and behaviour of the hyperreflective opacities were analysed individually using OCT image processing.

As a representation of total immune response [6,9], the total area of opacities/cells was quantified. In both SIG models, induced eyes showed significantly increased total areas (MsDx > MsDxF) versus healthy control animals (Figure 4a). To find out if the increase in total cell area was because of an increased number or cell size, and thus an increase in activated cells, the mean number of opacities was quantified over the study period. A constant number of opacities (10–20) was found in the healthy control cohort, in contrast to a higher and fluctuating number of opacities found in both SIG cohorts (approximately 45–35 opacities/cells in MsDx and MsDxF, respectively). Both SIG cohorts showed an initial increase, coinciding with the first intraocular injection and with OHT levels (Figure 4b). The results of these two in-depth analyses concur with previous findings [18,26].

**Figure 4 biomedicines-12-00633-f004:**
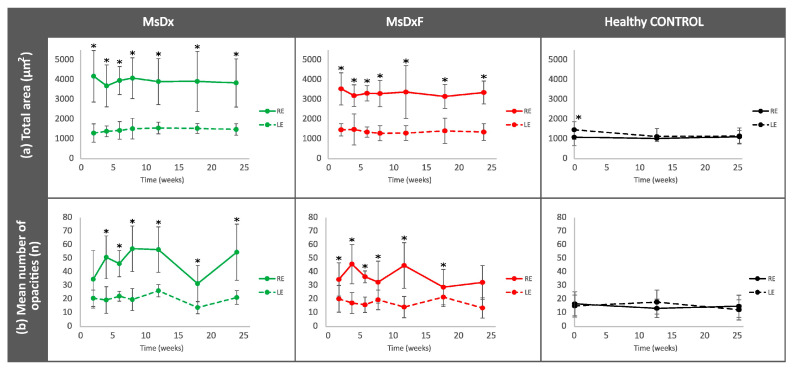
Changes in total immune response (**a**) and cellular quantification (**b**) in both steroid-induced glaucoma and healthy control animals. Abbreviations: RE: right eye; LE: left eye; MsDx: cohort with microspheres loaded with dexamethasone; MsDxF: cohort with microspheres loaded with dexamethasone and fibronectin injected into the anterior chamber; n: number; *: statistical significance (*p* < 0.05), using ANOVA test.

To assess the reproducibility and reliability of the measurement, the hyperreflective opacities or vitreous cell populations were divided, as we carried out previously in [18], into isolated cells (<10 µm^2^), non-activated cells (10–50 µm^2^), activated cells (50–250 µm^2^) and cell complexes (>250 µm^2^) [6,9] (Figure 5). This division based on size was possible because the study of the vitreoretinal interface does not require a correction factor and consequently can be measured directly.

Cell populations maintain similar sizes over time, implying reliability of measurement. Complexes > 250 µm^2^ undergo the biggest variations, with peaks at the onset of damage. Statistically significant differences (*p* < 0.05) were highlighted with alphabetic markers as follows: a (group 1–group 2), b (group 1–group 3), c (group 1–group 4), d (group 2–group 3), e (group 2–group 4) and f (group 3–group 4).

##### Percentage of Opacities/Cells by Size

Changes in the proportion in the non-activated and activated state in both SIG cohorts versus healthy eyes are shown in Figure 6. The healthy controls and both SIG cohorts showed a population ratio ordered from lowest to highest as follows: isolated cells (less than 10 µm^2^) < complexes (more than 250 µm^2^) < activated cells (50–250 µm^2^) < non-activated cells (10–50 µm^2^). A specular response was found between opacities of 10–50 µm^2^ (non-activated cells) and 50–250 µm^2^ (activated cells) and between opacities of 50–250 µm^2^ (activated cells) and opacities > 250 µm^2^ (complexes). Dynamic fluctuations were observed in both SIG cohorts, but on average, opacities 10–50 µm^2^ in size (non-activated cells) comprised approximately 40–50%. Both SIG cohorts maintained an anti-inflammatory profile throughout the study, with the MsDx model exhibiting a lower proportion of activated cells and higher cumulative intraocular dexamethasone release. This contrasts with the non-steroid glaucoma models, which had a higher percentage of opacities 50–250 µm^2^ in size (activated cells) [18].

**Figure 5 biomedicines-12-00633-f005:**
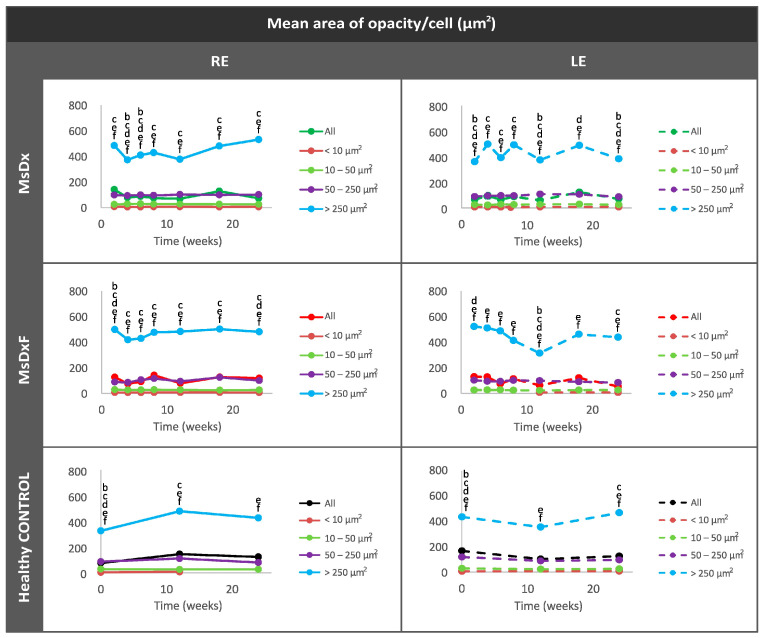
Cell subdivisions based on the mean area of vitreous opacities measured using OCT. Statistically significant differences (*p* < 0.05) were highlighted with alphabetic markers as follows: a (group 1–group 2), b (group 1–group 3), c (group 1–group 4), d (group 2–group 3), e (group 2–group 4) and f (group 3–group 4). Abbreviations: MsDx: cohort with microspheres loaded with dexamethasone; MsDxF: cohort with microspheres loaded with dexamethasone and fibronectin injected into the anterior chamber; isolated cells: < 10 µm^2^ (group 1); non-activated cells: 10–50 µm^2^ (group 2); activated cells: 50–250 µm^2^ (group 3); cell complexes: >250 µm^2^ (group 4).

**Figure 6 biomedicines-12-00633-f006:**
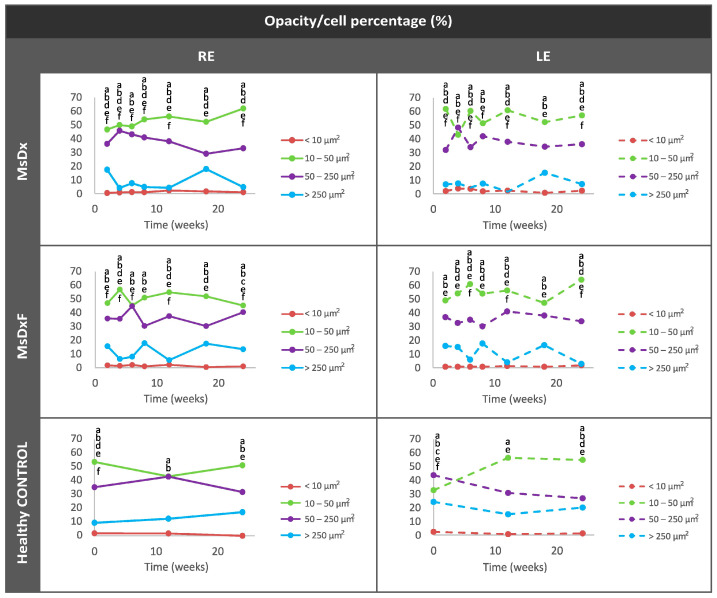
Changes in the vitreous immune population (opacities) in both steroid-induced glaucoma and healthy control animals throughout 6 months. Abbreviations: MsDx: cohort with microspheres loaded with dexamethasone; MsDxF: cohort with microspheres loaded with dexamethasone and fibronectin injected into the anterior chamber; isolated cells: opacities < 10 µm^2^ (group 1); non-activated cells: 10–50 µm^2^ (group 2); activated cells: 50–250 µm^2^ (group 3); cell complexes: >250 µm^2^ (group 4). Data represented as percentages. Statistically significant differences (*p* < 0.05) were highlighted with alphabetic markers as follows: a (group 1–group 2), b (group 1–group 3), c (group 1–group 4), d (group 2–group 3), e (group 2–group 4) and f (group 3–group 4).

##### Average Eccentricity of the Opacities/Cells

This analysis enhanced the characterisation of cell morphology as rounded morphology (eccentricity close to 0) versus linear or flat morphology (eccentricity close to 1). In healthy controls and both SIG cohorts, isolated opacities/cells (<10 µm^2^) presented the most rounded or amoeboid morphology (eccentricity 0.85) as opposed to opacities/cells with progressively larger sizes of 10–50 µm^2^ (non-activated), followed by those measuring 50–250 µm^2^ (activated cells) and <250 µm^2^ (cell complexes), these being increasingly flat (eccentricity 0.95–1). However, both SIG cohorts showed higher roundness in isolated cells (0.4–0.7) than in healthy cells (0.8) (Figure 7). In both SIG cohorts, the lower eccentricities coincided with increases in OHT in the MsDx model at week 4 (both sexes) (Figure 2a) and week 18 (in males) (Figure 2b), and in the MsDxF model at week 4 (both sexes) and week 24 (higher in females) (Figure 2c). The MsDx model with the highest IOP levels showed the lowest eccentricities at those times. The higher roundness of the isolated cells is related to recruitment to the noxa. Our findings were in accordance with a previous study showing that the number of intravitreal cells was higher in adult mice with experimentally elevated IOP [10].

**Figure 7 biomedicines-12-00633-f007:**
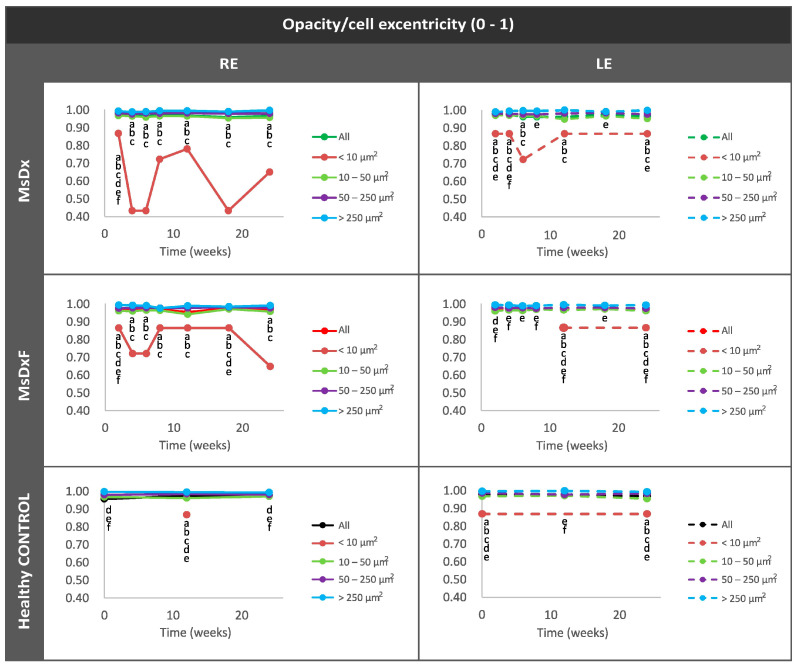
Mean eccentricity of vitreous opacity detected using in vivo OCT, according to size, in both steroid-induced glaucoma and healthy control animals. Indirect study of cell soma morphology. Abbreviations: MsDx: cohort with microspheres loaded with dexamethasone; MsDxF: cohort with microspheres loaded with dexamethasone and fibronectin injected into anterior chamber; isolated cells: opacities < 10 µm^2^ (group 1); non-activated cells: 10–50 µm^2^ (group 2); activated cells: 50–250 µm^2^ (group 3); cell complexes: >250 µm^2^ (group 4). Statistically significant differences (*p* < 0.05) were highlighted with alphabetic markers as follows: a (group 1–group 2), b (group 1–group 3), c (group 1–group 4), d (group 2–group 3), e (group 2–group 4) and f (group 3–group 4).

##### Mean Intensity of Opacities/Cells

Under physiological conditions, the lowest intensity was quantified in isolated opacities/cells (<10 µm^2^) and progressively increased with size: opacities of 10–50 µm^2^ (non-activated cells) followed by opacities of 50–250 µm^2^ (activated cells). However, in both SIG cohorts, the greatest change in intensity was quantified in the smallest opacities/cells (<10 µm^2^) (Figure 8) as a manifestation of activation of intracellular machinery and coinciding with the increase in size. As soma size increased (activated cells with pseudopod formation) [13,28,44,45], there was a relative decrease in mean intensity.

**Figure 8 biomedicines-12-00633-f008:**
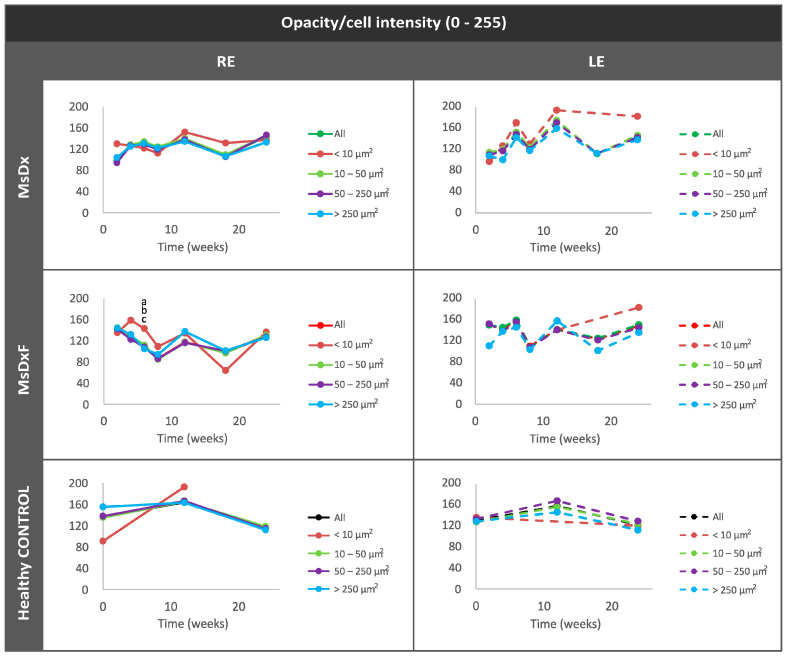
Mean intensity of opacities/cells based on size in both steroid-induced glaucoma and healthy control animals. Abbreviations: MsDx: cohort with microspheres loaded with dexamethasone; MsDxF: cohort with microspheres loaded with dexamethasone and fibronectin injected into the anterior chamber; isolated cells: opacities < 10 µm^2^; non-activated cells: 10–50 µm^2^; activated cells: 50–250 µm^2^; cell complexes: >250 µm^2^. Statistically significant differences (*p* < 0.05) were highlighted with alphabetic markers as follows: a (group 1–group 2), b (group 1–group 3) and c (group 1–group 4). Mean Orientation of the Opacities/Cells.

Orientation was analysed to measure an active shift (change in mean orientation) of immunity towards the damage [11,14,34,46]. The healthy control cohort did not experience any change. However, both SIG cohorts (MsDxF > MsDx) showed a change in orientation of the smallest opacities (<10 µm^2^: isolated ovoid cells) around 12 weeks (Figure 9), when both SIG cohorts experienced an increase in neuroretinal thickness, as found in our previous studies with these same models (Appendix A) [19,20].

**Figure 9 biomedicines-12-00633-f009:**
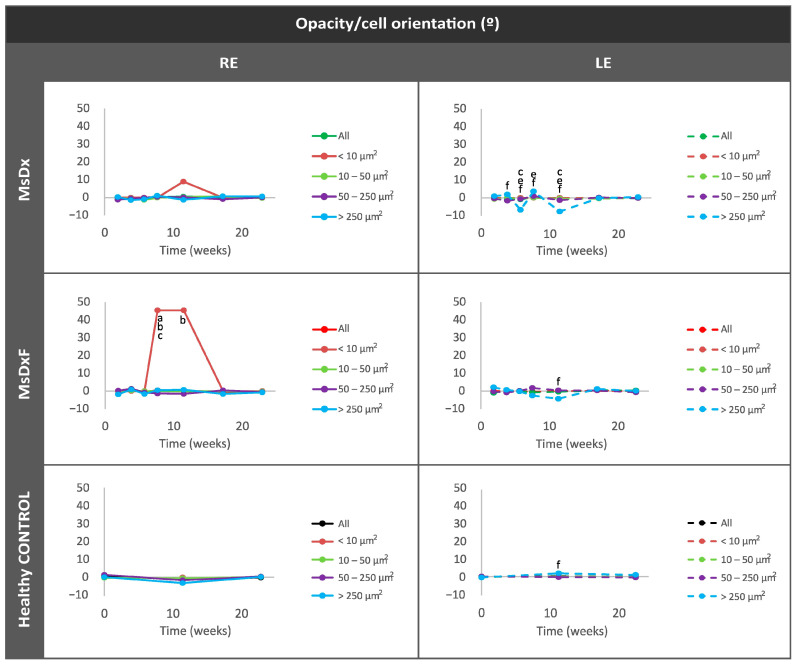
Mean orientation of vitreous opacity detected using OCT, according to size, in both steroid-induced glaucoma and healthy control animals. In vivo analysis for motility. Abbreviations: MsDx: cohort with microspheres loaded with dexamethasone; MsDxF: cohort with microspheres loaded with dexamethasone and fibronectin injected into the anterior chamber; isolated cells: opacities < 10 µm^2^ (group 1); non-activated cells: 10–50 µm^2^ (group 2); activated cells: 50–250 µm^2^ (group 3); cell complexes: >250 µm^2^ (group 4). Statistically significant differences (*p* < 0.05) were highlighted with alphabetic markers as follows: a (group 1–group 2), b (group 1–group 3), c (group 1–group 4), e (group 2–group 4) and f (group 3–group 4).

## 4. Discussion

Glaucoma is a multifactorial pathology in which immunity seems to be an early and important factor [47]. Elevated IOP triggers an innate immune response involving resident immune cells, such as microglia, and the infiltration of macrophages/monocytes and other secondary responses are responsible for RGC degeneration in glaucoma [7]. The primary response may be initially favourable in protecting the eye; it restores tissue equilibrium and promotes tissue cleaning, healing and functionality. If there is a defect in immune response pathways due to accumulating risk factors, prolonged and sustained or restrained inflammatory stimulation, or “neo-antigens” generated with ageing, the physiological homeostasis may be disrupted and the regulatory mechanisms are altered [48], thereby converting beneficial immunity into a neurodestructive autoimmune process [49]. Proinflammatory markers [50] and early cytokine dysregulation have been demonstrated independently and prior to the detection of RGC and axonal loss [7,51]. The subsequent events that lead to long-lasting activation of glial cells and adaptive immune responses become destructive, disrupting the homeostasis of the retina and resulting in the dysfunction of the immune-privileged status of the eye [3].

A possible therapeutic target for alleviating non-IOP-dependent factors could be modulating immunity. Controlling immune activation reduced optic nerve damage and reactive microglia-mediated neuroprotection in mouse retina, but eliminating it proved disadvantageous [7,52,53]. This study shows concordant results. Both SIG models showed lower vitreous signal intensities and lower counts of hyperreflective opacities (Figure 3 and Figure 4) than those found in previous non-steroid glaucoma models [18], but computational analysis of vitreous hyperreflective opacities revealed similar behaviour (Figure 6, Figure 7, Figure 8 and Figure 9). However, these SIG models developed significant neuroretinal damage with worse electroretinographic functionality, reduced structural thickness and lower RGC counts [19]. In other words, the presence of the steroid, which a priori could be thought to exert a protective anti-inflammatory effect, did not; rather, it produced and worsened the glaucomatous damage, as occurs in corticosteroid-induced glaucoma in humans [54]. Recent studies suggest that the immunity activated in glaucoma may not be counterbalanced by efficient immune suppression, and a greater stimulation response is characterised by increased proliferation and proinflammatory cytokine secretion. The potent anti-inflammatory effect of dexamethasone is well known. However, in this paper, it was used to generate two SIG models via sustained release from biodegradable Ms. Dexamethasone exerts a beneficial protective effect in a situation of overt and active inflammation [55], but in our study, we started from animals with no acute inflammation to counteract, sustainedly creating a potential imbalance in the delicate equilibrium of ocular immune privilege. Resident immunity may have been altered by overriding ocular immune inhibitors [56], and the balance tipped towards proinflammation with significant neuroretinal damage in both models.

Several research groups have tried to study immunity in vivo, but the need for genetically modified animals or the development of highly complex technology was prohibitive [48], and also ex vivo with dead animals as a result [57]. Enabled by the ability of light to pass through different optical densities, OCT is a fast, non-invasive device that provides in vivo scans of the neuroretinal structure and measurement of the retinal layers [58,59]. However, OCT has the handicap of not being able to differentiate among cell types within the neuroretinal thickness from glial, supporting or vascular cells. In the retina [9], in vivo microglia tracking using cSLO imaging has been reported [60]. This technique requires the use of animals genetically modified to express fluorescence, and thus, it cannot be used on humans. Analysis of the vitreoretinal interface using OCT, however, is nowadays a standard technique employed in ophthalmological clinics. Our group recently demonstrated that vitreous immune cells can be detected as hyperreflective opacities at the vitreoretinal interface and monitored using OCT imaging [18] in healthy and glaucomatous animals, coinciding with another group, who also confirmed it by confocal immunofluorescence in retinal vascular disease [61]. In the first phase, we focused on VIT/RPE relative intensity after a positive correlation with clinical vitreous turbidity was demonstrated [62] and validated under different conditions [17,26,63]. In the second phase, deeper computational analysis was performed to characterise the hyperreflective opacities that were confirmed by histology as hyalocyte-like Iba1+ cells (a microglial marker). Microglia and macrophages undergo characteristic morphological changes with their function. In an inactive situation, they are branched to sense changes in the microenvironment. The vitreous medium’s high water content makes it ideal for the transmission of soluble molecules, meaning that hyalocytes can easily and rapidly detect changes in the microenvironment [64] to target the noxa. Dexamethasone is a soluble molecule that was injected into the anterior chamber. It could target the vitreous chamber and modify the vitreous microenvironment, causing both SIG cohorts to maintain an anti-inflammatory profile since a higher proportion of non-activated cells were counted throughout the study (Figure 6). Furthermore, when damaged, hyalocytes [13] are activated and change shape (the number/size of intracellular organelles and, thus, the membrane content increase, facilitating detection by OCT imaging as hyperreflectivity), proliferate and migrate. These cells were associated with areas of retinal nerve fibre layer degeneration in glaucomatous patients [46], and a study using OCT detected a higher density of vitreous opacities close to areas of cell death [25].

Our previous paper on vitreous analysis in glaucoma detected increased intensity (onset of intracellular machinery) in the smallest opacities and a change in orientation (onset of displacement) as indicators of activation [18]. It suggested that the smallest opacities would be the first detected changes and could serve as an early marker of immune activation in the vitreous. This study of SIG corroborates these results. Eccentricity and intensity seem to be related to the increase in IOP (Figure 2*a and*
Figure 7), while the change in orientation seems to be related to the increase in retinal thickness (Figure 9 and Appendix A). Orientation is particularly relevant in the MsDxF cohort, which showed a marked change (Figure 9), indicating cells are oriented towards a certain point (retinal damage). This change in orientation coincided with an increase in the mean number of opacities (Figure 4b), suggesting that more cells were directed to the same retinal area. At later stages, the mean orientation returns to 0, which suggests that there could be more neuroretinal areas damaged in different areas and, therefore, more orientations to be adopted by the cells, nullifying the summation effect of the mean orientation. This hypothesis was confirmed by the results observed at later stages in the MsDx cohort. MsDx showed a further increase in cells (Figure 4b) but no change in orientation was observed, suggesting that these cells took several orientations and cancelled out their effect, which coincided with the increased neuroretinal damage evidenced by low RGC counts in our previous study [19]. It is possible that the return to 0 orientation of the opacities/hyalocytes in the two models could indicate “ageing or motility damage” because of the accumulation, repetition and chronicity of oxidative stress [35].

In this SIG study, the ocular barrier is altered in both Ms models by means of an intraocular injection in the anterior chamber that triggers an inflammatory response (ACAID response) [65]. In contrast to previous hypertensive models such as those induced by episcleral vein sclerosis or unloaded microspheres, no direct correlation was found between inducing injections, vitreous intensity or vitreous microglial or hyalocyte response [10]. This suggests that the pro-inflammatory effect of the intraocular injection was partly counteracted by the anti-inflammatory effect of the initial release of dexamethasone. As with non-SIG models [18], those animals with the lowest initial IOP had the highest IOP after the application of hypertensive stimuli, and in the MsDx cohort, higher IOP at week 2 correlated with higher vitreous signals detected by OCT at week 24 (r = 0.988, *p* = 0.002). Similarly, in left eyes, an inverse correlation was found between baseline IOP levels and vitreous OCT at week 12 (r = −0.851, *p* = 0.032), reflecting possible higher late contralateral inflammatory vitreous activation in the more hypotensive (susceptible) non-induced eyes. In this sense, different studies of glaucoma models have demonstrated stable, lower-intensity immune activation in the retina [66] and vitreous of the contralateral eye after OHT induction [18]. In this study, no increase in the number of opacities was observed in contralateral eyes, which means the hyalocytes were resident (without recruitment), but changes were detected in cell populations, eccentricity and intensity. Characterisation of hyperreflective opacities/immune cells (hyalocyte-like Iba1+ cells) in the healthy control cohort was explored in depth in our previous paper [18,61]. In summary, healthy rats showed a higher proportion of vitreous opacities in OCT, coinciding with sizes corresponding to non-activated cells, consistent with an anti-inflammatory or steady state [61] of the eye and immune inhibitory privilege [56,65]. A variable number of opacities corresponding to cells activated to maintain homeostasis was also quantified [11].

Limitations and future perspectives: There is still a long way to fully achieve the understanding of immune cell actions and cascades in both healthy and glaucomatous eyes. The authors are aware that there remain many aspects and unknowns to resolve. The obtained results could not be easily extrapolated to humans and more data are needed regarding the effectiveness of this study. The limitations in the histologic phenotyping of our study meant it could not clarify the paradigm of the infiltrative origin [67] of the increased vitreous opacities, although previous groups demonstrated the involvement of blood-derived inflammatory cells [61]. It would be beneficial to perform a computational study correlating the changes in the parameters of eccentricity, intensity and orientation of the vitreous opacities with neuroretinal changes, and to corroborate our topographic hypothesis of a change in orientation of those vitreous cells towards the RGC damage cluster by means of whole-mount histological studies or with new in vivo damage detection techniques [68,69]. Glaucoma therapies based on neuro-immunomodulatory targeting are emerging and immune cells could be important candidates [35].

## 5. Conclusions

Our method of analysing vitreous opacities/hyalocytes using OCT could serve as a promising imaging biomarker to detect immunity in the eye. It could help in the early diagnosis of disease onset and progression applicable to glaucoma and, potentially, in other multifactorial neurodegenerative diseases [70,71]. This study supports the previous evidence that simple, non-invasive in vivo analysis of glaucoma from the immune perspective is possible. We corroborate and validate the computational analysis of the individual hyperreflective opacities as a better technique than the overall relative measure of immunity using VIT/RPE by OCT. A clear example is the MsDxF cohort, which hardly exhibited any difference in vitreous signal intensity compared to the healthy control cohort, but in which subsequent detailed analysis showed higher cell counts and the study of eccentricity, intensity and orientation characteristics using OCT coincided with the clinical milestones of increased IOP or neuroretinal change. In view of the above, we believe that the individualised study of vitreous opacities is a reproducible and reliable method and offers information that can be correlated with clinical data, which could serve as a non-invasive biomarker in glaucoma diagnosis.

## Figures and Tables

**Figure 1 biomedicines-12-00633-f001:**
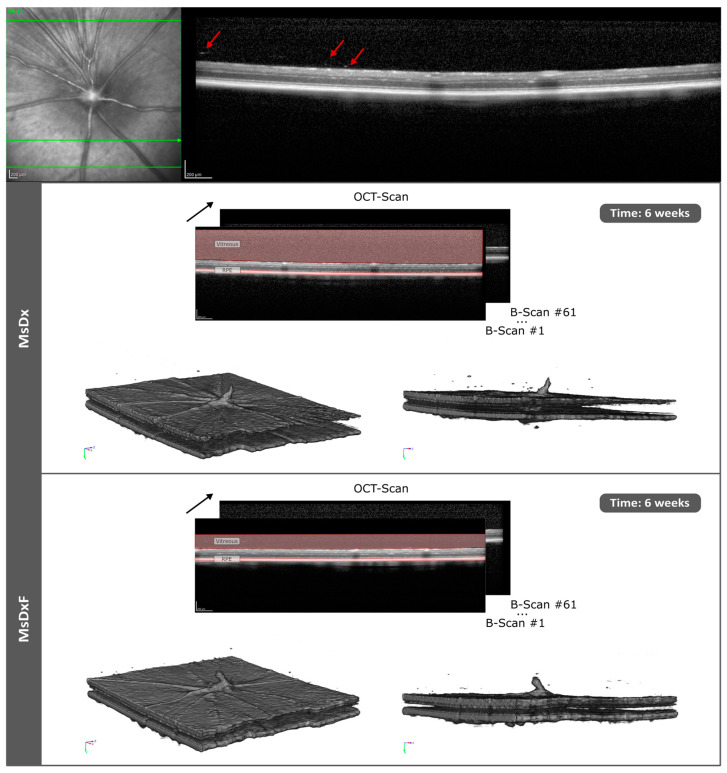
OCT scan and 3D reconstruction of 61 right-eye B-scans in two models of steroid-induced glaucoma at 6 weeks’ follow-up. The black arrow indicates image sequencing by optical coherence tomography (serial slices). Abbreviations: MsDx: cohort with microspheres loaded with dexamethasone; MsDxF: cohort with microspheres loaded with dexamethasone and fibronectin injected into the anterior chamber; OCT: optical coherence tomography; RPE: retinal pigment epithelium. Red arrows show the vitreous opacities.

**Table 1 biomedicines-12-00633-t001:** Correlations in both steroid-induced glaucoma and healthy control animals. Abbreviations: RE: right eye; LE: left eye; IOP: intraocular pressure; OCT: optical coherence tomography; w: weeks; MsDx: cohort with microspheres loaded with dexamethasone; MsDxF: cohort with microspheres loaded with dexamethasone and fibronectin injected into the anterior chamber; HC: healthy controls; im: inverse moderate correlation; m: moderate correlation. In bold: statistically significant correlations.

	Right Eye			Left Eye		
	MsDx	MsDxF	HC	MsDx	MsDxF	HC
**IOP/IOP**	4 w/6 w(m)	6 w/12 w (m)	**4 w/8 w** (r = 0.934, *p* = 0.020)	2 w/18 w (im)	2 w/4 w (m) 4 w/6-8-24 w (m)	
**IOP/OCT**	**2 w/24 w** (r = 0.988, *p* = 0.002) **4 w/24 w** (r = 0.896, *p* = 0.040)		**18 w/24 w** (r = 0.854, *p* = 0.031)	**4 w/18 w** (r = 0.889, *p* = 0.043)	**0 w/8 w** (r = 0.882, *p* = 0.020) **0 w/12 w** (r = −0.851, *p* = 0.032) **6 w/8 w** (r = 0.813, *p* = 0.049)	24 w/24 w (im)

## Data Availability

The authors confirm that the data supporting the findings of this study are available within the article and its Appendix A. Derived data supporting the findings of this study are available from the corresponding author on request.

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
