# Peer review of "Immune Analysis Using Vitreous Optical Coherence Tomography Imaging in Rats with Steroid-Induced Glaucoma"

_biomedicines, 2024, doi:10.3390/biomedicines12030633_

Round 1
Reviewer 1 Report
Comments and Suggestions for Authors
The paper Immune analysis using vitreous OCT imaging in rats 1 with steroid-induced glaucoma is within the scope of Biomedicines. It is well written and nicely organized paper. The results are clearly presented, the methodology is described in details. Some minor aspects should be addressed before the publication:
1. There is no clear aim of the study. At the end of the Introduction part the aim of the study should be declared.
2. It should be emphasized that the obtained results could not be easily extrapolated to humans and that more data ae needed regarding the safety and effectiveness of the study either in the limitation part or in the conclusion.
3. The paper should be formatted according to the requirements of the journal.
Author Response
The paper Immune analysis using vitreous OCT imaging in rats 1 with steroid-induced glaucoma is within the scope of Biomedicines. It is well written and nicely organized paper. The results are clearly presented, the methodology is described in details. Some minor aspects should be addressed before the publication:
- There is no clear aim of the study. At the end of the Introduction part the aim of the study should be declared.
Authors’ response: thank you for your suggestion. In the new version of the manuscript the aim of the study was included at the end of the introduction as follows:
This paper aims to corroborate the reliability of using computational OCT image analysis of hyperreflective opacities in the vitreous as a biomarker of vitreous immunity, in this case in two chronic steroid-induced glaucoma (SIG) rat models previously developed by our research group by injecting biodegradable microspheres (Ms) loaded with dexamethasone (MsDx) and a combination of dexamethasone and fibronectin (MsDxF) (with sustained release of the active compounds) into the anterior chamber of the eye.[19,20] Chronic exposure to glucocorticoids can raise IOP and is known to exert a negative effect in the form of maladaptive glial cell alterations and neuron damage or loss,[21] leading to SIG.[22] We corroborate and validate the computational analysis of the individual hyperreflective opacities as a better technique than the overall relative measure of immunity using OCT. The study of eccentricity, intensity and orientation characteristics of vitreous opacities using OCT is a reproducible and reliable method of non-invasive information in SIG.
- It should be emphasized that the obtained results could not be easily extrapolated to humans and that more data ae needed regarding the safety and effectiveness of the study either in the limitation part or in the conclusion.
Authors’ response: thank you for your suggestion. In the new version of the manuscript the limitation regarding extrapolation to humans was included.
- The paper should be formatted according to the requirements of the journal.
Authors’ response: now this new version was formatted according to Biomedicine requirements.
Reviewer 2 Report
Comments and Suggestions for Authors
Immune analysis using vitreous OCT imaging in rats 1 with steroid-induced glaucoma.
biomedicines-2881588
The authors studied the relationship between elevated immune response and glaucoma. the optical coherence tomography detects the immune response in the vitreous as hyperreflective opacities and these are subsequently characterized by computational analysis. This authors monitered the vitreous parainflammation in two steroid-induced glaucoma (SIG) animal models created with drug delivery systems (microspheres loaded with dexamethasone and dexamethasone/fibronectin). The study indicates that hyperreflective opacities in the vitreous using OCT could be a biomarker of glaucoma onset and progression.
The authors have presented thei research in a suystremati way. theintrduction is well defined, the results section is well organized and explained.
Majoer comments:
1. The authors should follow the templete suggested by the journal.
2. The figures images should be of high quality.
3. OCT images should be demonstrated.
4. The results section needs further explanation.
Comments on the Quality of English Language
Minor editing of English language required
Author Response
You can find our responses in the attached file
